# Multivariable association discovery in population-scale meta-omics studies

**Himel Mallick**[1,2], **Ali Rahnavard**[3], **Lauren J. McIver**[1,2], **Siyuan Ma**[1,2], **Yancong Zhang**[1,2], **Long H. Nguyen**[1,4,5], **Timothy L. Tickle**[2], **George Weingart**[1,2], **Boyu Ren**[1,2], **Emma H. Schwager**[1,2], **Suvo Chatterjee**[6], **Kelsey N. Thompson**[1], **Jeremy E. Wilkinson**[1], **Ayshwarya Subramanian**[1,2], **Yiren Lu**[1‡], **Levi Waldron**[7], **Joseph N. Paulson**[8], **Eric A. Franzosa**[1,2], **Hector Corrada Bravo**[9], **Curtis Huttenhower**[1,2]*

**1** Biostatistics Department, Harvard T. H. Chan School of Public Health, Boston, Massachusetts, United States of America, **2** The Broad Institute, Cambridge, Massachusetts, United States of America, **3** Computational Biology Institute, Department of Biostatistics and Bioinformatics, Milken Institute School of Public Health, George Washington University, Washington DC, United States of America, **4** Clinical and Translational Epidemiology Unit, Massachusetts General Hospital and Harvard Medical School, Boston, Massachusetts, United States of America, **5** Division of Gastroenterology, Massachusetts General Hospital and Harvard Medical School, Boston, Massachusetts, United States of America, **6** Epidemiology Branch, Division of Intramural Population Health Research, Eunice Kennedy Shriver National Institute of Child Health and Human Development, National Institutes of Health, Bethesda, Maryland, United States of America, **7** Department of Epidemiology and Biostatistics, CUNY School of Public Health, New York City, New York, United States of America, **8** Department of Biostatistics, Product Development, Genentech, Inc., South San Francisco, California, United States of America, **9** Center for Bioinformatics and Computational Biology, University of Maryland, College Park, Maryland, United States of America

‡ Unavailable
* chuttenh@hsph.harvard.edu

**Data Availability Statement:** The implementation of MaAsLin 2 is publicly available with source code, documentation, tutorial data, and as an R/ Bioconductor package at http://huttenhower.sph.

## Abstract

It is challenging to associate features such as human health outcomes, diet, environmental conditions, or other metadata to microbial community measurements, due in part to their quantitative properties. Microbiome multi-omics are typically noisy, sparse (zero-inflated), high-dimensional, extremely non-normal, and often in the form of count or compositional measurements. Here we introduce an optimized combination of novel and established methodology to assess multivariable association of microbial community features with complex metadata in population-scale observational studies. Our approach, MaAsLin 2 (Microbiome Multivariable Associations with Linear Models), uses generalized linear and mixed models to accommodate a wide variety of modern epidemiological studies, including cross-sectional and longitudinal designs, as well as a variety of data types (e.g., counts and relative abundances) with or without covariates and repeated measurements. To construct this method, we conducted a large-scale evaluation of a broad range of scenarios under which straightforward identification of meta-omics associations can be challenging. These simulation studies reveal that MaAsLin 2's linear model preserves statistical power in the presence of repeated measures and multiple covariates, while accounting for the nuances of meta-omics features and controlling false discovery. We also applied MaAsLin 2 to a microbial multi-omics dataset from the Integrative Human Microbiome (HMP2) project which, in addition to

harvard.edu/maaslin2. The software packages used in this work are free and open source, including bioBakery methods available via http://huttenhower.sph.harvard.edu/biobakery as source code, cloud-compatible images, and installable packages. Analysis scripts using these packages to generate figures and results from this manuscript (and associated usage notes) are available from https://github.com/biobakery/maaslin2_benchmark. The iHMP dataset is publicly available at the IBDMDB website (https://ibdmdb.org) and the HMP DACC web portal (https://www.hmpdacc.org/ihmp/). The processed HMP2 datasets analysed in this manuscript are also available as Supporting Information.

**Funding:** This work was funded in part by US National Science Foundation grant DEB-2028280 (AR), US National Institutes of Health grants U19AI110820 (CH, to Owen White), R01HG005220 (CH, to Rafael Irizarry), and R24DK110499 and U54DK102557 (CH). The funders had no role in study design, data collection and analysis, decision to publish, or preparation of the manuscript.

**Competing interests:** I have read the journal's policy and the authors of this manuscript have the following competing interests: CH is on the Scientific Advisory Board for Seres Therapeutics and Empress Therapeutics. The remaining authors have declared that no competing interests exist. Author Yiren Lu was unable to confirm their authorship contributions. On their behalf, the corresponding author has reported their contributions to the best of their knowledge.

reproducing established results, revealed a unique, integrated landscape of inflammatory bowel diseases (IBD) across multiple time points and omics profiles.

## Author summary

Recently, several statistical methods have been proposed to identify phenotypic or environmental associations with features (e.g., taxa, genes, pathways, chemicals, etc.) from molecular profiles of microbial communities. Particularly for human microbiome epidemiology, however, most of these are primarily focused on univariable associations that analyze only one or a few environmental covariates. This is a critical gap to address, given the growing commonality of population-scale microbiome research and the complexity of associated study designs, including dietary, pharmaceutical, clinical, and environmental covariates, often with samples from multiple time points or tissues. Surprisingly, there have been no systematic evaluations of statistical analysis methods appropriate for such studies, nor consensus on appropriate methods for scalable microbiome epidemiology. To this end, we developed and validated a statistical model (MaAsLin) that provides both the first unified method and the first large-scale, comprehensive benchmarking of multivariable associations in population-scale microbial community studies. We hope that the MaAsLin 2 implementation, validated through extensive simulations and an application to HMP2 IBD multi-omics, will be helpful for researchers in future analysis of both human-associated and environmental microbial communities.

This is a *PLOS Computational Biology* Software paper.

## Introduction

Human-associated microbiota has been convincingly linked to the development and progression of a wide range of complex, chronic conditions including inflammatory bowel diseases (IBD), obesity, diabetes, cancer, and cardiovascular disorders [1,2]. Due to recent advances in multiple high-throughput functional profiling technologies, research has expanded well beyond bacteria-specific 16S rRNA gene amplicon profiles to multi-omics surveys, i.e., non-bacterial, metagenomic, metatranscriptomic, metabolomic, and metaproteomic measurements assessed simultaneously in the same biological specimens [3,4]. Additionally, due to diminishing sequencing costs, longitudinal, within-subject study designs are becoming increasingly common, especially when assessing the microbiome in population health [5,6]. These large, complex data contain abundant information to enable microbe-, gene-, and compound-specific hypothesis generation at scale. However, robust quantitative methods to do so at scale can still be challenging to implement without excessive false positives—one of the main hurdles in accurate translational applications of the microbiome to human health.

One of the primary limitations of leveraging such population-wide multi-omics surveys is thus computational, in part due to the complexity and heterogeneity of microbial community data that have made reliable application of statistical methods difficult. In particular, best practices to guard against spurious discoveries in meta-omics datasets remain scarce [7–14]. High-

throughput meta-omics datasets have specific characteristics that complicate their analyses: high-dimensionality, count and compositional data structure, sparsity (zero-inflation), over-dispersion, and hierarchical, spatial, and temporal dependence, among others. To combat these challenges, specialized methods implemented in usable, reproducible software are needed to accurately characterize microbial communities within large human population studies, while maintaining both sensitivity and specificity.

Early advances in microbiome epidemiology focused on omnibus testing for identifying over-all associations between aggregate microbiome structure and host or environmental phenotypes and covariates (e.g., disease status, diet, antibiotics or medication usage, age, etc.). Many of these rely on permutation-based procedures for moderated significance testing [11]. These methods assess whether overall community patterns of variation are associated with the covariates of interest, but fail to provide feature-level inference to enable follow-up characterization (where a feature can be any profiled omics abundance, e.g., taxa, genes, pathways, chemicals, etc.) To facilitate actionable outcomes, it is important to discern feature-specific associations at the highest possible resolution. This has led to the development of a variety of per-feature (or feature-wise) association testing methods, most of which are based on similar statistical frameworks, differing primarily in (i) the choice of normalization or transformation, (ii) observation model or likelihood, and (iii) the associated statistical inference [11]. As compared to omnibus testing approaches, per-feature methods (i) identify associations for each individual feature-metadata pair, (ii) facilitate feature-wise covariate adjustment, and (iii) call out specific features (as opposed to complex combinations of features implicated in associations in omnibus testing), leading to increased interpretability for translational and basic biological applications.

Despite a rich literature on feature-wise association testing for microbial communities, methods that can accommodate a wide variety of modern epidemiological study designs remain scarce. For instance, many early methods do not explicitly account for the sparsity observed in microbial meta-omics observations, and only a few scale beyond routine univariate (differential abundance) analyses without becoming overly susceptible to false positive or false negative results [7,11]. Furthermore, most methods for microbiome data do not explicitly adjust for repeated measures and multiple covariates in a unified statistical framework, a lack of which can have a profound (and typically anti-conservative) impact on subsequent epidemiological inference.

Here, we address these issues by providing a flexible approach to identify multivariable associations in large, heterogeneous meta-omics datasets. We have implemented this method as MaAsLin 2 (Microbiome Multivariable Associations with Linear Models, with software version 2.0 released with this study), a successor to MaAsLin 1 [15,16]. Unlike MaAsLin 1's single-model framework based on applications of arcsine square root-transformed linear model following Total Sum Scaling (TSS) normalization [15,16], MaAsLin 2 has evaluated and combined the best set of analysis steps to facilitate high-precision association discovery in microbiome epidemiology studies. It provides a coherent paradigm through a multi-model framework with arbitrary coefficients (representing association strengths between phenotypes and covariates) and contrasts of interest, along with support for data exploration, normalization, and transformation options to aid in the selection of appropriate data- and design-driven statistical techniques for analyzing microbial multi-omics data. In this study, we also conducted a large-scale synthetic evaluation of a broad range of circumstances under which straightforward identification of meta-omics features can be challenging. These simulation studies revealed that MaAsLin 2 preserves statistical power in the presence of repeated measurements and multiple covariates while accounting for the nuances of meta-omics features and, critically, controlling false discovery rates. We concluded with an application to novel bio-marker discovery in multiple omics datasets from the Integrative Human Microbiome Project

(iHMP or HMP2 [6]). The implementation of MaAsLin 2, associated documentation and tutorial, and example data sets are freely available in the MaAsLin 2 R/Bioconductor software package at https://huttenhower.sph.harvard.edu/maaslin2.

## Design and implementation

MaAsLin 2 provides a comprehensive multi-model system for performing multivariable association testing in microbiome profiles—taxonomic, functional, or metabolomic—with analysis modules for preprocessing, normalization, transformation, and data-driven statistical modeling to tackle the challenges of microbial multi-omics (compositionality, overdispersion, zero-inflation, variable library size, high-dimensionality, etc.; **Fig 1A**). The MaAsLin 2 implementation requires two inputs: (i) microbial feature abundances (e.g., taxa, genes, transcripts, or metabolites) across samples, in either counts or relative counts; and (ii) environmental, clinical, or epidemiological phenotypes or covariates of interest (together "metadata"). Both metadata and microbial features are first processed for missing values, unknown data values, and outliers. If indicated, microbial measurements are then normalized and transformed to address variable depth of coverage across samples. Feature standardization is optionally performed, and a subset or the full complement of metadata is used to model the resulting quality-controlled microbial features and define p-values for each metadata association per feature using one of a wide range of possible multivariable models. After all features are evaluated, p-values are adjusted for multiple hypothesis testing and a table summarizing statistically significant associations is reported. While the default MaAsLin 2 implementation uses a log-transformed linear model on TSS-normalized quality-controlled data, the software supports several other statistical models including count models (e.g., Negative Binomial [17]), zero-adjusted models (e.g., Compound Poisson [18–20], Zero-inflated Negative Binomial (ZINB) [21]), and multiple normalization/transformation schemes under one estimation umbrella. In the presence of repeated measures, MaAsLin 2 additionally identifies covariate-associated microbial features by appropriately modeling the within-subject (or -environment) correlations in a mixed model paradigm, while also accounting for inter-individual variability by specifying between-subject random effects in the model. A variety of summary and diagnostic plots are also provided to visualize the top results.

## Results

### MaAsLin2 validation

To identify model components appropriate for MaAsLin 2's microbiome per-feature association testing and to objectively benchmark current association methods, we assessed realistic synthetic datasets generated by SparseDOSSA [22,23] (full details of individual association methods, as well as simulation parameters, are described in **S1** and **S2 Texts** and are available online at https://github.com/biobakery/maaslin2_benchmark). Briefly, SparseDOSSA is a synthetic data generation routine that models biologically plausible synthetic data from diverse template microbiome profiles by considering (i) feature-feature, (ii) feature-metadata, and (iii) metadata-metadata correlations, superseding previous efforts by including multiple covariates and longitudinal designs (**S1 Text**). As compared to previous simulation schemes, SparseDOSSA allows multivariable spike-in both in the presence and absence of repeated measures, as well as arbitrary covariance structure in the metadata design matrix.

For this study, we carried out several spike-in experiments to induce and test controlled associations, as governed by configurable simulation parameters (**S1 Fig**). When used for this purpose, SparseDOSSA first generates null microbial community features containing no significant association patterns using a Bayesian hierarchical model independently of metadata

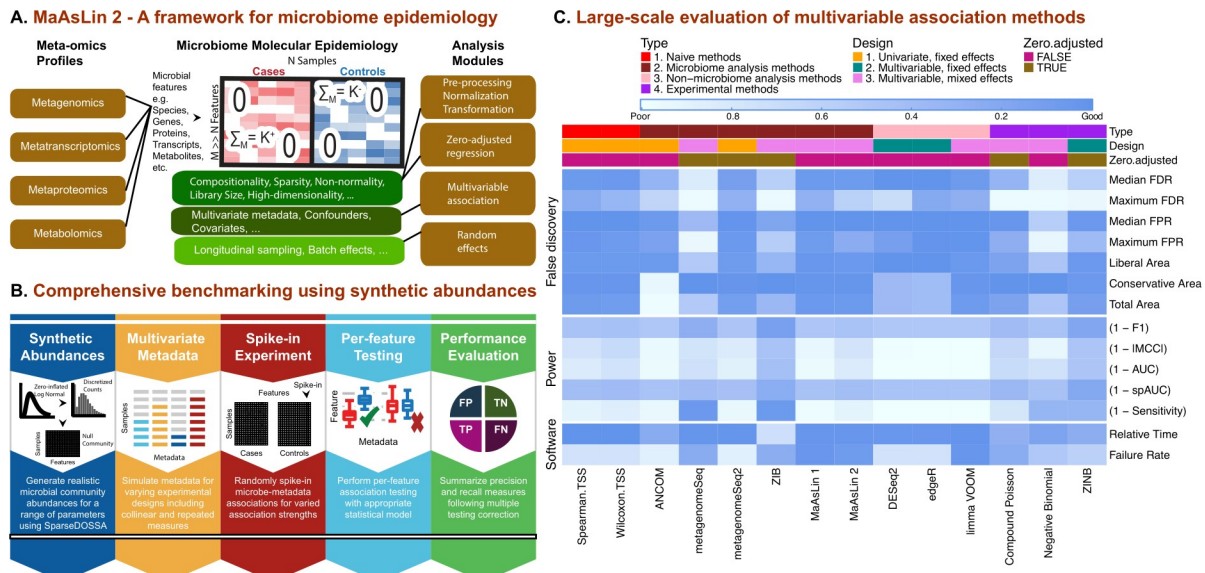

**Fig 1. MaAsLin 2 for feature-wise association of microbial communities with phenotypes. A**) MaAsLin 2 is a statistical method for association analysis of microbial community meta-omics profiles. It comprises several steps, including data transformation, multivariable inference, multiple hypothesis test correction, and visualization. These are based on a set of flexible and computationally efficient linear models, while accounting for the nuances of microbiome data, repeated measures, and multiple covariates. **B)** Comprehensive benchmarking of multivariable methods for microbiome epidemiology. To identify appropriate methods for associating microbiome features with health outcomes and other covariates, we assessed up to 84 combinations of normalization/transformation, zero-inflation, and regression models (S1A Fig). These were applied to synthetic data using a hierarchical model (SparseDOSSA, http://huttenhower.sph.harvard.edu/sparsedossa) to generate realistic, model-agnostic datasets with varying scopes and effect sizes of microbiome associations. Individual per-feature association methods were performed repeatedly to evaluate method-specific recall and precision measures. **C**) Association method performance summary across major evaluation criteria. Three aspects of performance were considered: (i) false discovery, (ii) sensitivity, and (iii) computational efficiency. Evaluation metrics (S1B Fig) are shown (in rows) for the resulting microbial multivariable association methods (both state-of-the-art and novel), averaged over all simulation parameters (S1A Fig). The top-performing methods (as measured by average F1 score) from each class of models (S1C Fig) are shown (in columns). Except for Spearman and Wilcoxon that maintained best performance on TSS-normalized data, all methods exhibited superior performance with no/default normalization (ANCOM, metagenomeSeq, metagenomeSeq2, DESeq2, edgeR, MaAsLin 1, MaAsLin 2, limma VOOM, ZIB) or library size normalization in which log-transformed library size is included as an offset in the associated GLM likelihood (Compound Poisson, Negative Binomial, ZINB). Top colored boxes represent method characteristics including the capability to handle zero-inflation and random effects. Based on synthetic evaluations, MaAsLin 2 includes optimized default models for epidemiological testing in microbial multi-omics data.

features (**Fig 1B** and **S1 Text**). In addition to varying sample size and feature dimension, a broad range of metadata and experimental designs are then considered, including repeated measures and univariate and multivariate covariates (both continuous and binary) of varying dimension and effect size (**S1A Fig**). Specifically, in each instance, we varied sample sizes from small (10) to large (200) for a fixed feature size (up to 500), and within each sample size, the effect size parameter was again varied from modest (e.g., <2-fold differences) to strong (10-fold). In each simulation, 10% of features (and 20% of metadata for multivariable scenarios) were modified as an in-silico spike-in (**S1 Text**). Precision and recall measures (**S1B Fig**) were averaged over 100 simulation runs. All methods were corrected for multiple hypothesis testing using standard approaches for FDR control, declaring significant associations at a target of FDR 0.05. For a fair comparison, a basic, model-free filtering step to remove low-abundance features was performed before statistical modeling for all methods (**S2 Text**). Methods unable to process specific simulation configurations due to high computational overhead or slow convergence were omitted for those cases.

To compare the detection power of various methods in identifying true positive feature associations, we first comprehensively evaluated published metagenomic tools and

representative methods from bulk RNA-seq literature within each simulation scenario. These methods were combined with several microbiome-appropriate normalization, transformation, and linkage models (**S1C Fig** and **S2 Text**). In particular, we considered six distinct categories of methods in our evaluations: (i) published methods specifically designed for microbial communities, such as metagenomeSeq [24], ANCOM [14,25], and ZIB [26,27], (ii) published bulk RNA-seq differential expression methods, such as DESeq2 [28], edgeR [29], and limma VOOM [30,31]; (iii) existing generalized linear model (GLM) count models, such as the negative binomial [17], (iv) methods based on linear models, such as limma [32] and "pure" linear models (LMs); (v) representative zero-adjusted methods from the microbiome and single-cell RNA-seq literature such as the Compound Poisson [18–20] and the ZINB [21,33]; and finally (vi) traditional, simplistic nonparametric methods, such as Spearman correlation and Wilcoxon tests. Of note, many of these methods can only compare two groups (i.e., a single binary metadatum), and not all are compatible with continuous and multivariate metadata, resulting in a distinct set of comparable methods for each experimental design.

Our first consideration in designing MaAsLin 2 for microbiome epidemiology was to ensure that both current statistical theory and practical issues were considered during the analysis of microbiome multi-omics data. To this end, we rigorously characterized finite-sample properties of various association methods focusing on three broadly defined aspects: (i) false discovery, (ii) detection power, and (iii) software implementation, with multiple performance indicators for each category (**Figs 1C** and **S1B**). Rather than focusing on a single evaluation metric like the Area Under the Curve (AUC) or the False Positive Rate (FPR), we ranked methods based on a combination of metrics (**S1B Fig** and **S2 Text**), many not considered in previous benchmarking. To summarize each evaluation criteria, a normalized continuous score ranging between 0 and 1 was assigned (**S2 Text**). Methods were then eliminated based on the presence of 'red flags' with respect to at least one evaluation criteria, i.e., extreme departure from the best possible value. Finally, metrics that are mainly descriptive rather than quantitative were also evaluated (e.g., the ability to handle complex metadata designs, zero-inflation, or repeated measures) to achieve a final consensus. For simplicity, we thus abbreviate any extreme departure from a metric's best possible value as a 'red flag'. This tiered strategy not only allowed us to select a set of "best" methods based on the fewest 'red flags' across all scenarios, but also to identify a method that is (i) sufficiently robust to false discovery control and detection power, (ii) scalable to large multi-omics datasets, and (iii) accommodating of most modern epidemiological designs and microbial data types.

Notably, previous benchmarking in this area has only focused on differential abundance testing without the simultaneous consideration of multiple covariates and repeated measures [7–9]. Additionally, with the exception of Hawinkel et al. [7], these benchmarking efforts lacked important considerations to the extent that they either (i) did not consider FDR as a metric of evaluation [9,34,35] or (ii) misreported false positive rate as FDR [8] (**S2 Text**). While most of these studies made a final recommendation based on the traditional AUC metric or a combination of sensitivity and specificity, we argue that without considering the FDR-controlling behavior of a method, the AUC values alone are too optimistic to draw any meaningful conclusions about its practical utility. In other words, particularly for biological follow-up, high precision among the most confident (lowest recall) predictions is essential. To this end, our large-scale benchmarking enables a progressive unification of traditional and practically important evaluation metrics by providing a comprehensive connected view of microbiome multivariable association methods, especially in the context of modern study designs, multiple covariates, and repeated measures.

Overall, our simulation study revealed that virtually all high-sensitivity methods with an overoptimistic median AUC, especially those targeted to microbial communities, exhibited a

highly inflated average FDR (**Fig 1C**, full results in **S1**–**S8 Data**). A similar pattern was observed for other AUC-like measures such as F1 score and Matthew's correlation coefficient (MCC). On the other end of the spectrum, compositionality-corrected methods such as ANCOM exhibited an extreme departure from 'good' performance with respect to several criteria including sensitivity and p-value calibration, as measured by both Conservative and Total Area [7] (**S2 Text**). Overall, these simulations reveal that while there is no single method that outperforms others in all scenarios, MaAsLin 2 was the only multivariable method tested that controlled FDR with the fewest 'red flags' across scenarios (**Fig 1C**).

This initial phase of our study thus expands the understanding of statistical association methods appropriate for microbial community data under varying study designs, and it especially highlights the inability of many common methods to control false discoveries. This is of critical importance to past and present microbiome association methods, as failure to control the FDR causes uncertainty in both scientific reproducibility and interpretability. Based on these evaluations, a linear model with TSS normalization and log transformation was adopted as the default model in MaAsLin 2, and the software provides several flexible options to apply a combination of other normalization, transformation, and statistical methods to customize specific analysis tasks. The implementation of MaAsLin 2, associated documentation, and example data sets are freely available both as an R/Bioconductor package and a command-line interface tool at https://huttenhower.sph.harvard.edu/maaslin2.

## MaAsLin 2 controls false discovery rate while maintaining power in differential abundance analysis

Differential abundance testing for microbial community features (taxa, pathways, chemicals, etc.) is one of the most commonly used strategies to identify features that differ between sample categories such as cases and controls. Despite a large number of developments in the area, a lack of consensus on the most appropriate statistical method has been a major concern [11]. To model experimental designs of this type, we used synthetic count data with spiked-in features differentially abundant between two defined groups of samples. In particular, we multiplied the mean relative abundance of a randomly sampled fraction of 10% of the features with a given effect size (fold change) in one of the groups and renormalized the ensemble of relative abundances to a unit sum to create features differentially abundant between groups. We repeated this procedure for each unique combination of sample size (10, 20, 50, 100, 200), feature dimension (100, 200, 500), and fold change (1, 2, 5, and 10), each time summarizing performance over 100 simulation runs (**S1 Text**). Before model fitting, features with a low prevalence (<10%) were trimmed from the generated data sets.

As in our overall evaluation (**Fig 1C**), we observed marked differences between the FDR-controlling behavior of different methods in the simple case of single binary metadatum and non-longitudinal design, in some cases exceeding 75% (**Fig 2**). Among the methods with good, robust FDR control, only those based on linear models achieved moderate power, whereas, for methods such as DESeq2 and edgeR, the FDR control came at the cost of reduced power. Among other methods, practically all count and zero-inflated models, as well as newer methods based on log-ratios such as ANCOM, struggled to correctly control the FDR at the intended (nominal) level, and the best performance in this class of methods was obtained by metagenomeSeq2, Compound Poisson, and ZINB (as measured by the F1 score). Many of the remaining methods were too liberal, with metagenomeSeq and Negative Binomial standing out with many false positive findings. Overall, linear models (LMs) remained critically the only class of methods tested that has good control of FDR across study designs, and they all

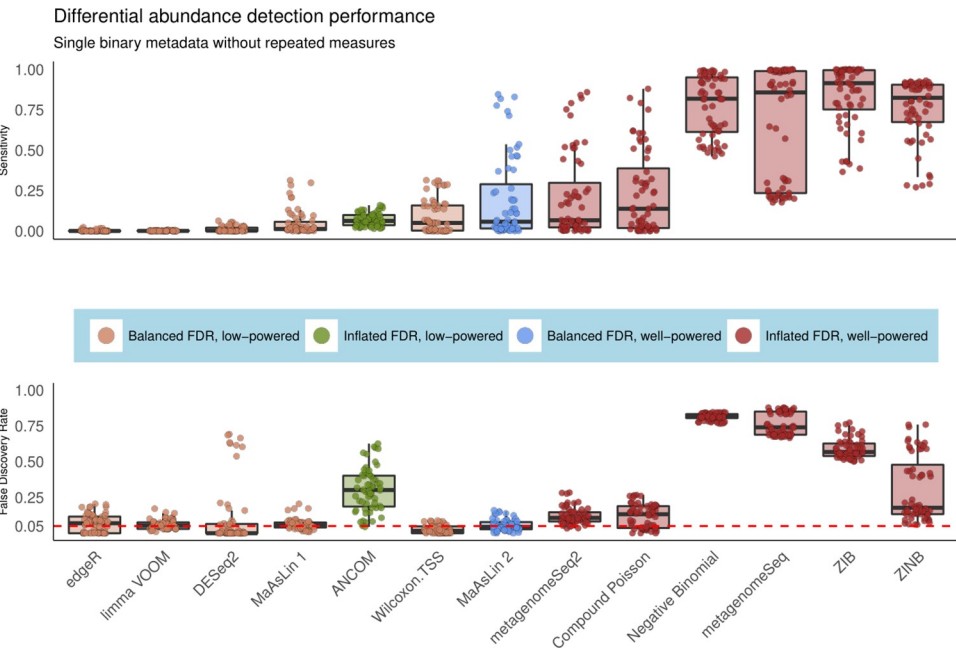

**Fig 2. MaAsLin 2 controls false discovery rate while maintaining power in differential abundance analysis of microbial communities.** To assess models' behaviors during differential abundance analysis, we simulated 100 independent datasets per parameter combination, each containing a single binary metadatum and a fixed number of true positive features (10% of features differentially abundant) for varying association strengths and sample sizes (**S1A Fig**). We then evaluated the ability of different microbiome association methods to recover these associations using a variety of performance metrics and summarized the results across runs. Both sensitivity and false discovery rates (FDR) are shown for the best-performing method from each class of models (as measured by average F1 score). Compared to zero-inflated and count-based approaches, MaAsLin 2's linear model formulation consistently controlled false discovery rate at the intended nominal level while maintaining moderate sensitivity (full results in **S1–S8 Data**). Red line parallel to the x-axis is the target threshold for FDR in multiple testing. Methods are sorted by increasing order of average F1 score across all simulation parameters in this setting.

exhibited a boost in statistical power with increased sample size and association strength (**S2 Fig**).

We also evaluated the average FPR of these methods by recording the fraction of tested unassociated (negative) features that were deemed significant following significance testing. Nearly all methods controlled the FPR well below the imposed level (**S3 Fig**). Relatedly, we employed a previously proposed metric called "departure from uniformity" (i.e., departure from a uniform distribution of p-values under the null), which, complementary to FPR, quantifies the liberal or conservative area (**S2 Text**) between observed and theoretical quantiles of a uniform distribution [7]. As expected, methods with high average false discovery rates, including zero-inflated and count models, showed extreme departures from uniformity in the liberal direction, whereas conservative methods such as DESeq2 and edgeR showed the same in the opposite direction, suggesting extreme violation of uniformly distributed p-values under the null hypothesis (**S4 Fig**). While these results raise potential concerns about the FDR-controlling behaviors of most existing methods, LM-based approaches did not exhibit this trend. In general, tools based on linear models (such as limma) performed very similarly when calibrated with MaAsLin 2's default model parameters, as expected, but not with their recommended default parameters (**S2–S4 Figs**). Additionally, their options for handling sparsity and compositionality were generally not appropriate for microbiome data. Amplicon, metagenomic taxonomic, and functional profiles each show distinct count and zero-inflation

properties, for example, that are best handled by a multi-model system. In addition to the binary metadata design, we repeated the above simulation experiments for univariate continuous metadata as well, which led to similar conclusions (**S5 Fig**), further supporting MaAsLin 2's default model's performance across metadata types and experimental designs.

As a final evaluation, we assessed the impact of various normalization schemes on the associated statistical modeling, evaluating all combinations of normalizations appropriate for each applicable method (**S1C Fig** and **S2 Text**). Focusing on the best-performing linear models, we found that model-based normalization schemes such as relative log expression (RLE [36]) as well as data-driven normalization methods such as the trimmed mean of M-values (TMM [37]) and cumulative sum scaling (CSS [24]) led to good control of FDR, but they also led to a dramatic reduction in statistical power (**S2**–**S5 Figs).** In contrast, TSS showed the best balance of performance among all tested normalization procedures, leading to more powerful detection of differentially abundant features. These results have potential implications for other analyses in addition to differential abundance testing, as normalization is usually the first critical step before any analysis of microbiome data, and an inappropriate normalization method may severely impact post-analysis inference. In summary, our synthetic evaluation indicates that TSS normalization, although simplistic in nature, may be superior to other normalization schemes especially in the context of feature-wise differential abundance testing (and more generally for multivariable association testing, as described later), in addition to community-level comparisons as previously described [38].

## MaAsLin 2 facilitates multivariable association discovery in population-scale epidemiological studies

Moving beyond univariate comparisons, we next assessed MaAsLin 2's performance in multivariable association testing in comparison to other multivariable methods. Although widespread in microarray and gene expression literature, multivariable analysis methods have remained underdeveloped in microbial community studies. From an epidemiological point of view, coefficients from a covariate-adjusted regression model are arguably more interpretable than its individual, unadjusted counterparts. As a result, major conclusions from existing benchmarking studies geared towards univariate associations are not generalizable to this broader setting, where challenges such as zero-inflation and multiple testing are likely to be exacerbated, especially in relation to multiple rounds of independently conducted univariate analyses as commonly practiced.

To introduce multivariable associations into synthetically generated microbial feature profiles, we supplemented each "sample" with multiple covariates consisting of both binary and continuous metadata, either independent or correlated (**S1A Fig** and **S1 Text**). In each of these datasets, 10% randomly selected features were modified ("spiked") to be associated with randomly chosen 20% metadata features with a given magnitude (effect size). After spiking in, samples were rescaled to their original (simulated) sequencing depth. As before, we repeated this procedure for each unique combination of sample size (10, 20, 50, 100, 200), feature dimension (100, 200, 500), and effect size (1, 2, 5, 10), each time summarizing performance over 100 simulation runs.

The results from this set of simulations revealed that MaAsLin 2's default linear model had the highest sensitivity among the methods that controlled the FDR at the target level, which also remained consistent at larger sample sizes and stronger effect sizes (**Fig 3**). We also observed an improvement in performance when TSS normalization was employed (as compared to no normalization) but did not observe similar improvement for other normalization methods (**S6 Fig**). As before, zero-inflated and count models failed to control the FDR at the

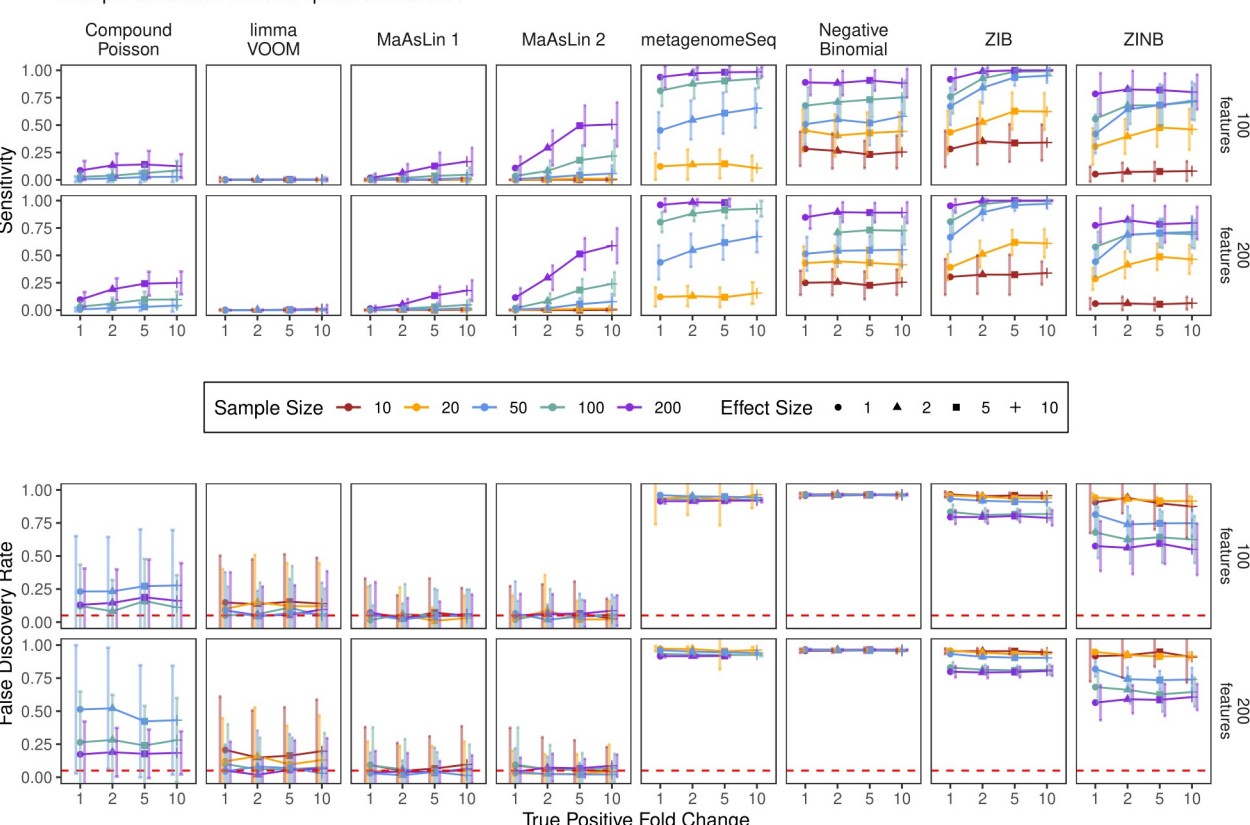

**Fig 3. MaAsLin 2 facilitates multivariable association discovery in large-scale human epidemiological and other microbial community studies.**
Synthetic datasets containing five "metadata" with varying types of induced feature associations were analyzed using a variety of multivariable approaches (**S1C Fig**). As measured by power (recall) and false discovery rate (FDR), MaAsLin 2's default linear model outperformed other methods in controlling FDR while maintaining power across true-positive fold-change values, regardless of the total number of features. As expected, MaAsLin 2 has better power for stronger effect sizes, eventually attaining the highest power among all FDR-controlling methods (full results in **S1**–**S8 Data**). Red line parallel to the x-axis is the nominal FDR. Values are averages over 100 iterations for each parameter combination. The x-axis (effect size) within each panel represents the linear effect size parameter; a higher effect size represents a stronger association. For visualization purposes, the best-performing methods from each class of models (as measured by average F1 score) are shown. Methods are sorted by increasing order of average F1 score across all simulation parameters in this setting.

nominal level, in the sense that the actual FDR was always above the nominal threshold used for identifying significant features—a phenomenon that was surprisingly consistent regardless of the metadata covariance structure (**S7 Fig**). Taken together, these findings further confirm that MaAsLin 2's default linear model is able to detect relevant associations across a broad range of metadata designs, facilitating population-level analyses of microbial communities.

## MaAsLin 2 enables targeted microbiome hypothesis testing in the presence of repeated measures

To further validate MaAsLin 2 for longitudinal (or other repeated measures) microbiome data, we modified our simulation scheme to introduce subject-specific random effects—a key feature of modern microbiome population studies [39]. To this end, we tested MaAsLin 2 and related methods on two types of study designs. The first comprised univariate binary metadata designed to be challenging by the inclusion of non-independence of the data across time points. Second, we also simulated more realistic datasets using multiple independent

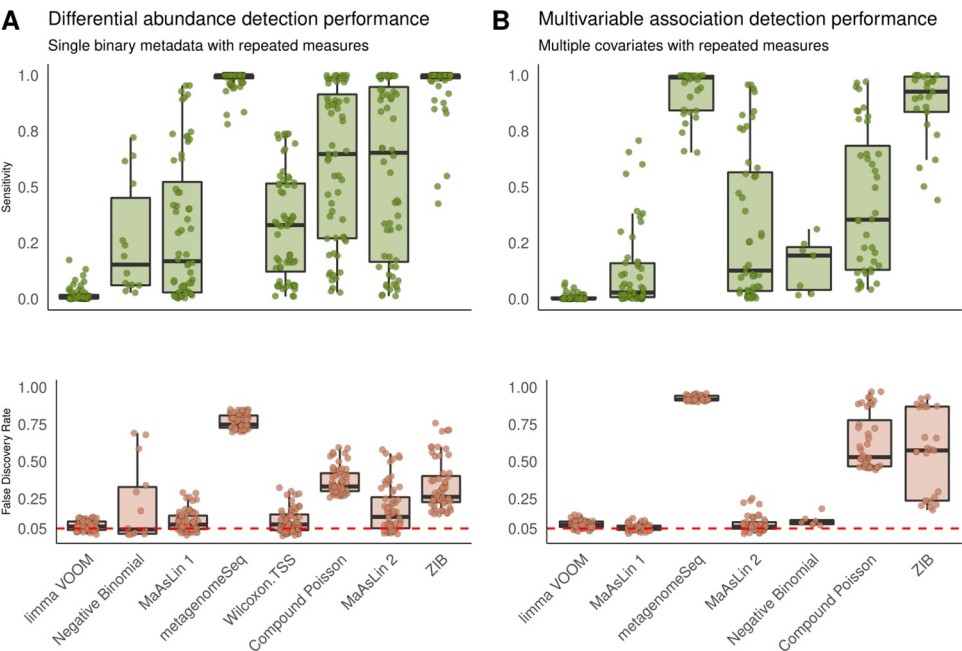

**Fig 4. MaAsLin 2 enables targeted microbial feature testing in the presence of repeated measures.** Results on simulated data comprising SparseDOSSA-derived compositions with five repeated measures per sample. The FDR is close to the target 0.05 level for MaAsLin 2's default linear model but not for zero-inflated and count models (full results in **S1–S8 Data**). As before, MaAsLin 2's linear model is consistently better powered than both negative binomial and limma VOOM at comparable FDR values, which remains consistent for both univariate continuous metadata (**A**) and multivariable metadata designs (**B**). The red line parallel to the x-axis is the given threshold for FDR in multiple testing. Within each panel, methods are sorted by increasing order of average F1 score across all associated simulation parameters in each setting.

covariates specific to longitudinal microbiome studies. In both these regimes, realistic data were generated using SparseDOSSA each with five time points, as in previous studies [27], but after introducing within-subject correlations and between-subject random effects drawn from a multivariate normal distribution (**S1 Text**). It is to be noted that the set of evaluable models is greatly reduced from the previous set of cross-sectional association tests, as methods not capable of assessing repeated measures were discarded.

Using these longitudinal synthetic "microbial communities," we compared the estimation and inference from MaAsLin 2 with those of the existing methods, which revealed that MaAsLin 2 had much lower false discovery rates than alternatives including ZIB (**Figs 4** and **S8–S11**), a method specifically designed for microbiome longitudinal data. Both ZIB and MaAsLin 2's linear mixed effects models are capable of identifying covariate-associated features by jointly modeling all time points. However, the computational overhead of ZIB is significantly higher than that of MaAsLin 2, which is prominent even for small datasets (**S12 Fig**). Notably, although not nearly as severe as count-based and zero-inflated models, MaAsLin 2 had a slightly inflated FDR in the univariate repeated measures scenario (**Fig 4A**) but not in the multivariable scenario (**Fig 4B**). Among other methods, methods based on generalized linear mixed models (GLMMs) such as Negative Binomial and Compound Poisson performed similarly to their non-longitudinal counterparts for both normalized and non-normalized counts (**S8 and S9 Figs**). This remained consistent for both univariate continuous metadata (**S10 Fig**) as well as multiple, correlated covariates (**S11 Fig**). Overall, these results suggest that MaAsLin 2's linear mixed effects model consistently provides lower false discovery rates across metadata

designs and can effectively aid in testing differential abundance and multivariable association of longitudinal microbial communities.

## Multi-omics associations from the Integrative Human Microbiome Project

We applied MaAsLin 2 to identify relevant microbial features associated with the inflammatory bowel diseases (IBD) using longitudinal multi-omics data from the Integrative Human Microbiome Project (iHMP or HMP2 [39]). The HMP2 Inflammatory Bowel Disease Multi-omics (IBDMDB) dataset included 132 individuals recruited in five US medical centers with Crohn's disease (CD), ulcerative colitis (UC), and non-IBD controls, followed longitudinally for one year with up to 24 time points each (**S3 Text**).

Integrated multi-omics profiling of the resulting 1,785 stool samples generated a variety of data types including metagenome-based taxonomic profiles as well as metagenomic and metatranscriptomic functional profiles, producing one of the largest publicly available microbial multi-omics datasets. Metagenomes and metatranscriptomes were functionally profiled using HUMAnN 2 [40] to quantify MetaCyc pathways [41], and taxonomic profiles from metagenomes were determined using MetaPhlAn 2 [42] (**S3 Text**). For each of these data modalities (i.e., taxonomic profiles and DNA/RNA pathways), independent filtering was performed before downstream testing to reduce the effect of zero-inflation on the subsequent inference. In particular, features for which the variance across all samples was very low (below half the median of all feature-wise variances) or with >90% zeros were removed [39]. To further remove the effect of redundancy in pathway abundances (explainable by at most a single taxon), only features (both DNA and RNA) with low correlation with individual microbial abundances (Spearman correlation coefficient <0.5) were retained.

We first used the IBDMDB to perform an additional semi-synthetic evaluation of association methods' performance in "real" data, specifically when attempting to associate randomized, null microbial taxonomic profiles to covariates (**S3 Text**). To this end, we permuted all samples 1,000 times to construct shuffled "negative control" datasets, each time assessing the number of significant associations (unadjusted p <0.05) for each applicable method. These were averaged across iterations to derive the expected number of null associations per method (which should remain near-zero under usual circumstances). In particular, we fit (i) a baseline model as a function of IBD diagnosis (a categorical variable with non-IBD as the reference group) while adjusting for enrollment age (as a continuous covariate) and antibiotic use (as a binary covariate), and (ii) a mixed effects model (with subject as random effects) with IBD dysbiosis state as an additional time-varying covariate in addition to the time-invariant covariates considered in the baseline model. Consistent with prior simulations, we found that several methods produced inflated empirical type I error rates (**S13 Fig**). This conclusion remained unchanged across varying significance thresholds, and as a result, we did not further apply these methods to the non-permuted data. Relevantly and importantly, linear models did not suffer from this problem, providing additional support for MaAsLin 2's robustness to false positive findings.

To dissect dysbiotic changes in IBD at greater resolution, we applied MaAsLin 2 to each individual microbial feature type (i.e., species and DNA/RNA pathways) to test association with IBD phenotype while controlling for IBD dysbiosis state, diagnosis, age, and antibiotic use (**Fig 5** and **S3 Text**). Nominal p-values for UC- and CD-specific effects were subjected to multiple hypothesis testing correction using the Benjamini-Hochberg method [43] with an FDR threshold of 0.25. MaAsLin 2 identified a comparable number of significant associations with those initially reported by the IBDMDB [39]. Among microbial species, MaAsLin 2's default linear model identified 222 significant associations (**S9 Data**), among which 134

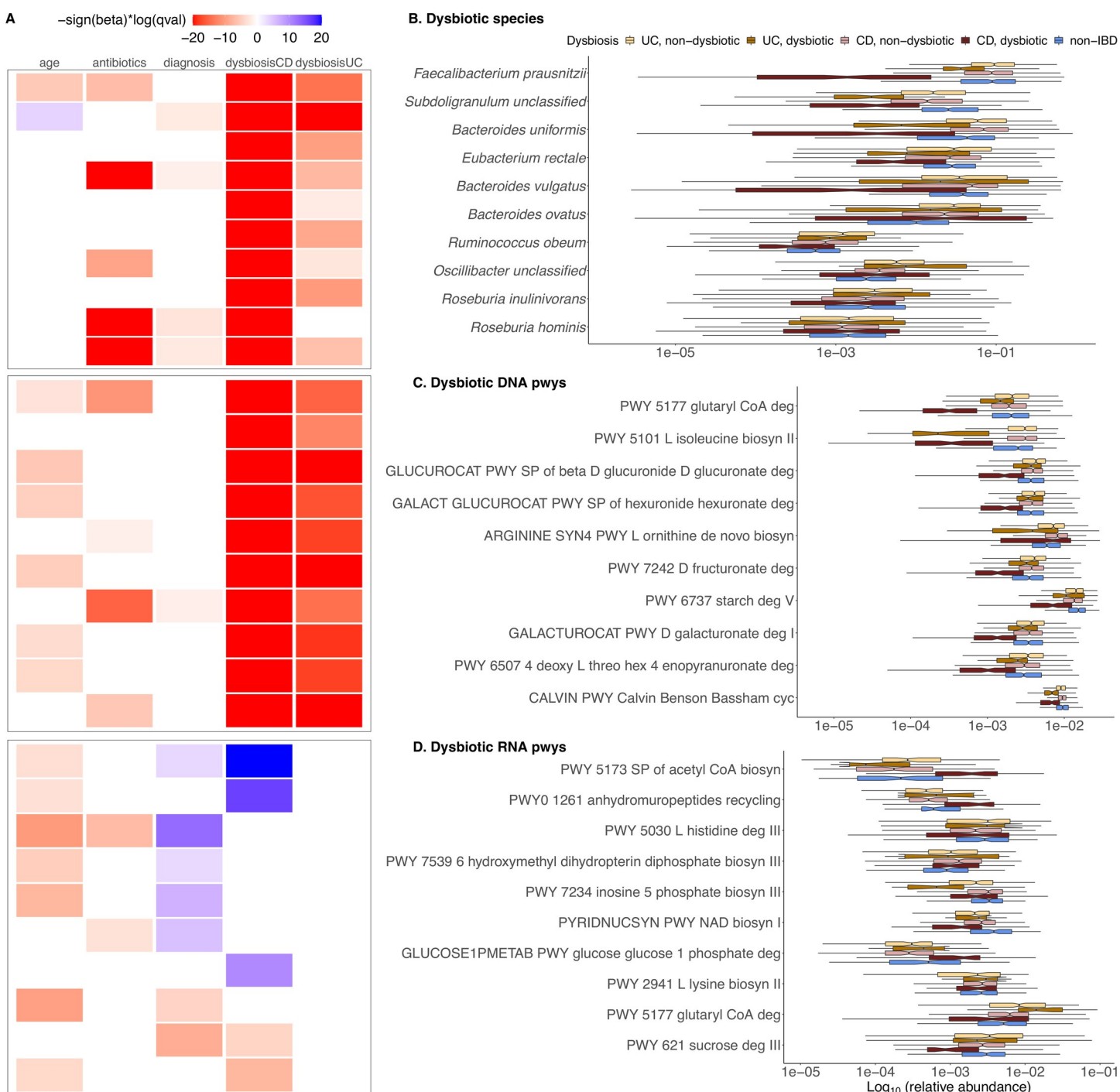

**Fig 5. Multi-omics associations from the Integrative Human Microbiome Project. A)** Top 10 significant associations (FDR < 0.25) detected by MaAsLin 2's default linear model (full results in S9–S14 Data). All detected associations are adjusted for subjects and sites as random effects and for other fixed effects metadata including the subject's age, diagnosis status (CD, UC, or non-IBD), disease activity (defined as median Bray-Curtis dissimilarity from a reference set of non-IBD samples), and antibiotic usage. **B,C,D)** Representative significant associations with dysbiosis state from each omics profile are shown: species (**B**), metagenomic (DNA) pathways (**C**), and metatranscriptomic (RNA) pathways (**D**). Values are log-transformed relative abundances with half the minimum relative abundance as pseudo count.

(60.4%) overlapped with the original study. MaAsLin 2 also reported many significant associations that were not discovered in the original study (**S14 Fig**). For instance, we observed a significant increase in *Bacteroides ovatus* in both UC and CD dysbiotic patients that was not previously captured, as well as detecting (with MaAsLin 2's increased power) specific depleted *Roseburia* species (*R. inulinivorans* and *R. hominis*) not captured by the previous analysis. Notably, top hits from both MaAsLin 2 and the original study yielded nearly identical rankings across data types, which broadly manifested as a characteristic increase in facultative anaerobes at the expense of obligate anaerobes, in agreement with the previously observed depletion of butyrate producers such as *Faecalibacterium prausnitzii* in IBD (**Fig 5A**).

As an additional validation, we next re-analyzed the HMP2 taxonomic and functional profiles using a zero-adjusted model (implemented in MaAsLin 2 as the Compound Poisson [18–20]). While this maintained type I error control in our shuffled data validation (as did the default linear model, **S13 Fig**)**,** it was generally less desirable due to FDR inflation in simulations (**Figs 2–4**). In terms of the number of differentially abundant features detected, both the default linear model and the Compound Poisson model performed similarly, with a significant overlap between the top hits identified by each method (**S15 Fig**). Among other methods, ZIB and limma VOOM also maintained good Type I error control in these experiments (**S13 Fig**), although again both underperformed along other axes in our simulation studies. These results further strengthen the finding that a combination of controlled parametric simulations and 'negative control' experiments based on data shuffling are useful together in identifying methods for real-world applications, as the lack of either can lead to misleading (and irreproducible) conclusions across independent evaluations [7]. This also highlights the flexibility of MaAsLin 2's multi-analysis framework, wherein researchers are well-served with multiple (i) normalization schemes, (ii) statistical models, (iii) multiplicity adjustments, (iv) fixed and random effects specifications, and (v) in-built visualization and pre-processing options, facilitating seamless application of methods across diverse experimental designs under a single estimation umbrella.

Finally, in addition to taxonomic associations, MaAsLin 2 also detected 399 and 58 significant functional associations for metagenomic (DNA) and metatranscriptomic (RNA) pathways, respectively (**S10** and **S11 Data**), among which 358 (89.7%) and 39 (67.2%) overlapped with the original study. While the original analysis of these data included only community-wide functional profiles, we extended this by considering metagenomic and metatranscriptomic functional profiles at both whole-community and species-stratified levels to quantify overall dysbiotic functions while simultaneously assigning them to specific taxonomic contributors. In particular, this considers a per-feature DNA covariate model [44], in which per-feature normalized transcript abundance is treated as a dependent variable, regressed on per-feature normalized DNA abundances along with other regressors in the model (**S3 Text**). Surprisingly, bioinformatics and statistics for metatranscriptomics are not yet standardized, and our results indicate that subtle model variations can produce substantially different results, due to the interactions between two compositions: DNA and RNA relative abundances (**S12 Data**)**.** This novel modeling strategy thus led to the discovery of several novel transcript associations relative to the original study (**S13** and **S14 Data**).

In many of these pathways, functional perturbations were driven by shifts in their characteristic contributing taxa (**Fig 5B**). For example, the most significant DNA pathways enriched in CD patients were characteristic of facultative anaerobes such as *Escherichia coli*, which are broadly more abundant during inflammation. These included pathways such as synthesis of the enterobactin siderophore, lipid A, and sulfate reduction. A second set of enriched pathways was depleted due to the loss of microbes such as *F. prausnitzii*, a particularly prevalent

organism that, when abundant, tended to contribute most of all enriched pathways it encodes in this cohort (e.g., synthesis of short-chain fatty acids and various amino acids).

With the increased sensitivity of this analysis for species-stratified pathways, the overwhelming majority of significant metagenomic differences were attributable solely to the most differential individual organisms, as expected (**S13** and **S14 Data**). Essentially every pathway reliably detectable in *E. coli* was enriched during CD, UC, or both, and most *F. prausnitzii* pathways depleted, along with many pathways from other gut microbes common in "health" (*Bacteroides vulgatus*, *B. ovatus*, *B. xylanisolvens*, *B. caccae*, *Parabacteroides* spp., *Eubacterium rectale*, several *Roseburia* spp., and others). Interestingly, since both more potentially causal "driver" pathways, along with all other "passenger" pathways encoded by an affected microbe, are detected by this more sensitive stratified analysis, it can be in many ways more difficult to interpret than the non-stratified, community-wide, cross-taxon metagenomic responses to broad ecological conditions such as immune activity, gastrointestinal bleeding, or oxygen availability.

Conversely, differentially abundant microbe- and pathway-specific transcript levels highlighted a much more specific and dramatic shift toward oxidative metabolism, away from anaerobic fermentation, and towards Gram-negative (often *E. coli*) growth during inflammation (**Fig 5C**) [45]. Many of these processes were either more extreme during (e.g., gluconeogenesis) or unique to (e.g., glutathione utilization) active CD, as compared to UC. CD and UC responses were opposed in a small minority of cases (e.g., glutaryl-CoA degradation). When stratified among contributing taxa, these differences were almost universally attributable to a few key species, particularly an increase in *E. coli* activity during inflammation and decreases of *F. prausnitzii* transcript representation. Condition-specific transcriptional changes were occasionally contributed (or not) by "passenger" *Bacteroides* spp. (*B. fragilis*, *B. xylanisolvens*, *B. dorei*) instead. Note that these differences include pathways more likely to be "causal" in some sense, as significant transcriptional changes were generally a subset of those detected due to whole-taxon shifts in DNA content (including housekeeping pathways such as general amino acid or nucleotide biosynthesis). These findings further support the importance of disease-specific transcriptional microbial signatures in the inflamed gut relative to metagenomic profiles of functional potential, suggesting that a potential loss of species exhibiting altered expression profiles in disease may have more far-reaching consequences than suggested by their genomic abundances alone.

## Availability and future directions

Limitations of the current MaAsLin 2 method include, first, its restriction to associating one feature at a time. While this strategy has the advantage of being straightforward to interpret, implement, and parallelize, it sacrifices inferential accuracy by ignoring any correlation structure among features. This can certainly exist due both to compositionality and to biology and will differ e.g., between taxonomic features (related by phylogeny) vs. functional ones (such as pathways). A potential extension would be to adopt an additional multivariate framework that allows modeling simultaneously rather than sequentially, thus improving power by borrowing strength across non-independent features. Second, as revealed by our synthetic evaluation, not surprisingly, linear models remain underpowered in detecting weak effects among microbial communities, especially when accompanied by a small sample size. This is in some ways a necessary consequence of the restrictions of current microbiome measurement technologies, and it emphasizes the importance of an informed power analysis before study planning to ensure an optimal sample size with adequate detection power. Third, it is not possible to capture the full range of differential biases and errors introduced by various bioinformatics pipelines using

a single, representative template dataset, as considered here. To this end, multiple, diverse taxonomic and functional template datasets can be considered for future benchmarking, potentially in combination with other upstream simulation frameworks such as CAMISIM [46] to investigate the effect of sequence assembly, genome binning, batch effects, taxonomic binning, taxonomic profiling, and other steps on differential analysis performance. Fourth, while we have focused on linear associations in this study, non-linear associations may also be of interest (as in other types of molecular epidemiology). Finally, and relatedly, it is not straightforward to incorporate any type of graph structure knowledge such as phylogeny or pathway-based functional roles into the per-feature linear model framework. Bayesian linear models can potentially improve on this by including such information through a suitable prior distribution.

Several aspects of microbiome epidemiology remain to be investigated both biologically and computationally, in addition to the challenges addressed here. For example, while it is possible to obtain strain-level resolution from metagenomic sequencing data, strain variants are generally unique to individuals rather than broadly carried by many people, presenting unique challenges for strain-level multi-omics. From a computational point of view, this calls for further refinements to MaAsLin 2's methodology when applied to strain-resolved community profiles. In addition, the modeling framework adopted here can only inform undirected associations, and hence cannot be used to infer causation. Advanced methods from other molecular epidemiology fields such as causal modeling and mediation analysis methods can help overcome these issues [47]. Another opportunity for future extension of our method is the integration of established missing data imputation methods across features and metadata, a common pitfall in many molecular epidemiology studies [39]. Combined, such extensions will lead to further improvement in downstream inference, allowing researchers to investigate a range of hypotheses related to differential abundance and multivariable association.

## Discussion

A longstanding goal of microbial community studies, be they for human epidemiology or environmental microbiomes, is to identify microbial features associated with phenotypes, exposures, health outcomes, and other important covariates in large, complex experimental designs. This parallels other methods for high-throughput molecular biology, but microbial community multi-omics must account for properties such as variable sequencing depth, zero-inflation, overdispersion, mean-variance dependency, measurement error, and the importance of repeated measures and multiple covariates. To this end, we have developed and validated a highly flexible, integrated framework utilizing an optimized combination of novel and well-established methodology, MaAsLin 2. This accommodates a wide variety of modern study designs ranging from within-subject, longitudinal to between-subject, cross-sectional, diverse covariates, and a range of quality control and statistical analysis modules to identify statistically significant as well as biologically relevant associations in a reproducible framework. The embedding of these strategies in the paradigm of generalized linear and mixed models enables the treatment of both simple and quite complex designs in a unified setting, improving the power of microbial association testing while controlling false discoveries. To validate this framework, we have extensively evaluated its performance alongside a set of plausible methods for differential abundance analysis in a wide range of scenarios spanning simple univariate to complex multivariable with varying scopes and effect sizes of microbiome associations. Finally, we applied MaAsLin 2 to identify disease-associated features by leveraging the HMP2's multi-omics profiles of the IBD microbiome, confirming known associations and suggesting novel ones for future validation.

A unique aspect of our synthetic evaluation of microbial community feature-wise association methods while developing MaAsLin 2 is their comprehensive assessment in the presence of multiple covariates and repeated measures, an increasingly common characteristic of modern study designs. To identify covariate-associated microbial features from longitudinal, non-independent measurements, it is necessary to jointly model data from all time points and appropriately account for the within-subject correlations while allowing for multiple covariates. This is particularly critical in the human microbiome, where baseline between-subject differences can be far greater than those within-subjects over time, or of the effects of phenotypes of interest. To the best of our knowledge, the synthetic evaluation presented here is the first to consider such aspects of large-scale microbiome epidemiology in statistical benchmarking. This enabled us to investigate key aspects of published methods that would be difficult to generalize from univariate comparisons alone [7–9]. Note that the resulting conclusion is largely independent of the association models being evaluated, as the synthetic data was generated from an additional, completely external model (i.e., the zero-inflated log-normal, **S1 Text**), which is fundamentally different from any of the evaluated parametric models. Our simulation results thus complement the findings of previous studies in several important aspects. Consistent with previous reports, nearly all zero-inflated models suffer from poor performance (i.e., inflated false positives and higher computation costs), here in both univariate and multivariable scenarios with or without repeated measures. This calls for methodological advancements in statistical modeling of zero-inflated data, as existing theory seems to differ very surprisingly from practice when implemented by established optimization algorithms and applied to noisy data.

One noteworthy finding of our evaluation is that a random effect implementation of the same underlying statistical model can lead to different substantive conclusions than its fixed effects counterpart. This was particularly evident for the negative binomial case, where a substantially better control of FDR (albeit inflated) was observed for the random effects analog. Interestingly, the negative binomial model (with or without zero-inflation) is in many ways considered the most "appropriate" model for count-based microbial community profiles, but we observed extremely inconsistent behavior for existing negative binomial and ZINB implementations during our evaluation, as also observed in previous findings [48]. In particular, our negative binomial evaluation used the *glm.nb()* function from the *MASS* R package [49] for fixed effects and the *glmer()* function from the R package *lme4* [50] for random effects, whereas the ZINB evaluation used the *zeroinfl()* function from the R package *pscl* [51]. This additionally highlights the potential reproducibility concerns induced by differences in algorithms, implementations, and computational environments even for the same underlying model, suggesting that great caution should be taken when interpreting multiple implementations of the same statistical model for challenging microbial community settings in the absence of an experimentally validated gold standard.

In agreement with previous studies, we confirmed that most RNA-seq differential expression analysis tools tend to provide suboptimal performance when applied unmodified to zero-inflated microbial community profiles. Count-based models, due to their strong parametric assumptions (i.e., parametric specifications of the mean-variance relationship), tend to have inflated FDR when the assumptions are violated. In sharp contrast to previous claims, however, compositionality-corrected methods such as ANCOM [14,25] as well as specialized normalization and transformation methods such as CLR [52] did not improve performance over non-compositional approaches [8,53], consistent with recent findings that compositional methods may not always outperform non-compositional methods [35]. Importantly, these conclusions hold regardless of the nature of the modeling paradigm (i.e., univariate vs. multivariable), thus providing a generalizable benchmark for future evaluation studies of applied

microbiome association methods. Though we primarily focused on data generated in microbial community surveys, many of our conclusions are extendible to similar zero-inflated count data arising in other applications such as single-cell RNA-seq. Taken together, these simulation results revealed that further investigation into the causes of the failure of FDR correction and development of specialized false positive-controlling methods are important upcoming challenges in microbiome statistical research.

As currently implemented, MaAsLin 2 is designed to be applicable to most human and environmental microbiome study designs, including cross-sectional and longitudinal. Clearly, these can also be extended to additional designs, such as nested case-control and case-cohort. It is to be noted that MaAsLin 2's capability extends well beyond association analysis. For instance, MaAsLin 2's multi-analysis framework has been used in the context of meta-analysis [54], and the extracted residuals and random effects from a MaAsLin 2 fit can be used for further downstream analysis (e.g., as has been done in the original HMP2 study for cross-measurement correlation analysis [39]). By adhering to a flexible mixed effects framework, MaAsLin 2 can analyze multiple groups and time points jointly with other associated covariates, which allows formulation of both fixed effects (for cross-sectional associations) and random effects (for within-subject correlations) in a single unified framework. This is particularly appropriate for non-longitudinal studies (those with a small number of repeated measures, e.g., multiple tissues or families), or from sparse and irregular longitudinal data from many subjects (e.g., with unequal number of repeated measurements per subject, as commonly encountered in population-scale epidemiology). This aspect could also be extended in the future, based on the increasing availability of dense time-series profiles appropriate for non-linear trajectory-based methods from Bayesian nonparametrics, such as Gaussian processes, particularly in the presence of multiple covariates [5,55]. Finally, methods need to be developed to accommodate the increasing availability of microbiome-host interactomics and electronic health records in population-scale microbiome-wide epidemiology, moving beyond observational discovery toward translational applications of the human microbiome. In summary, the methodology presented here provides a starting point for more efficient identification of microbial associations from large microbial community studies, offering practitioners a wide set of analysis strategies with state-of-the-art inferential power for the human microbiome and other complex microbial environments.

## Supporting information

**S1 Text. Data for differential feature model evaluations.** Descriptions of how the synthetic datasets are generated using SparseDOSSA for both univariate and multivariable metadata designs (with or without repeated measures) and the associated spike-in procedure to introduce feature-metadata associations.
(DOCX)

**S2 Text. Multivariable association test evaluation.** Details on how each of the methods compared in the **Results** section are implemented, run on the simulated data, and evaluated using various performance metrics.
(DOCX)

**S3 Text. Analysis of the iHMP (HMP2) IBDMDB multi-omics dataset.** Details on differential abundance analysis of iHMP (HMP2) IBDMDB multi-omics dataset using MaAsLin 2, along with a description of the associated study design, quality control procedures, shuffle data experiments, and per-feature multivariable models for various microbial measurement types.
(DOCX)

**S1 Data. Full summary of detection performance in synthetic benchmarking for single binary metadatum (UVB) without repeated measures.** Detection performance measures for all methods (after ignoring incompatible combinations) as averages over 100 iterations are provided for single binary metadatum design (UVB) without repeated measures (**S1** and **S2 Texts**).
(XLSX)

**S2 Data. Full summary of detection performance in synthetic benchmarking for single continuous metadatum (UVA) without repeated measures.** Detection performance measures for all methods (after ignoring incompatible combinations) as averages over 100 iterations are provided for single continuous metadatum design (UVA) without repeated measures (**S1** and **S2 Texts**).
(XLSX)

**S3 Data. Full summary of detection performance in synthetic benchmarking for multiple independent metadata (MVA) without repeated measures.** Detection performance measures for all methods (after ignoring incompatible combinations) as averages over 100 iterations are provided for multiple independent metadata design (MVA) without repeated measures (**S1** and **S2 Texts**).
(XLSX)

**S4 Data. Full summary of detection performance in synthetic benchmarking for multiple correlated metadata (MVB) without repeated measures.** Detection performance measures for all methods (after ignoring incompatible combinations) as averages over 100 iterations are provided for multiple correlated metadata design (MVB) without repeated measures (**S1** and **S2 Texts**).
(XLSX)

**S5 Data. Full summary of detection performance in synthetic benchmarking for single binary metadatum (UVB) with repeated measures.** Detection performance measures for all methods (after ignoring incompatible combinations) as averages over 100 iterations are provided for single binary metadatum design (UVB) with repeated measures (**S1** and **S2 Texts**).
(XLSX)

**S6 Data. Full summary of detection performance in synthetic benchmarking for single continuous metadatum (UVA) with repeated measures.** Detection performance measures for all methods (after ignoring incompatible combinations) as averages over 100 iterations are provided for single continuous metadatum design (UVA) with repeated measures (**S1** and **S2 Texts**).
(XLSX)

**S7 Data. Full summary of detection performance in synthetic benchmarking for multiple independent metadata (MVA) with repeated measures.** Detection performance measures for all methods (after ignoring incompatible combinations) as averages over 100 iterations are provided for multiple independent metadata design (MVA) with repeated measures (**S1** and **S2 Texts**).
(XLSX)

**S8 Data. Full summary of detection performance in synthetic benchmarking for multiple correlated metadata (MVB) with repeated measures.** Detection performance measures for all methods (after ignoring incompatible combinations) as averages over 100 iterations are provided for multiple correlated metadata design (MVB) with repeated measures (**S1** and **S2**

**Texts**).
(XLSX)

**S9 Data. MaAsLin 2 associations between HMP2 multi-omics features (metagenomic species) and covariates.** List of statistically significant associations (FDR<0.25) between species profiles and IBD disease phenotype (with non-IBD as reference), IBD dysbiosis state (with non-dysbiotic as reference), age, and antibiotic use using MaAsLin 2's default multivariable linear mixed effects model with subject and site as random effects (**S3 Text**). Features are sorted by minimum FDR-adjusted p-values. For each feature, coefficient estimates and test statistics and the associated two-tailed p-values are also reported. Input features and metadata are also provided.
(XLSX)

**S10 Data. MaAsLin 2 associations between HMP2 multi-omics features (unstratified DNA pathways) and covariates.** List of statistically significant associations (FDR<0.25) between unstratified DNA pathways and IBD disease phenotype (with non-IBD as reference), IBD dysbiosis state (with non-dysbiotic as reference), age, and antibiotic use using MaAsLin 2's default multivariable linear mixed effects model with subject and site as random effects (**S3 Text**). Features are sorted by minimum FDR-adjusted p-values. For each feature, coefficient estimates and test statistics and the associated two-tailed p-values are also reported. Input features and metadata are also provided.
(XLSX)

**S11 Data. MaAsLin 2 associations between HMP2 multi-omics features (unstratified RNA pathways) and covariates.** List of statistically significant associations (FDR<0.25) between unstratified RNA pathways and IBD disease phenotype (with non-IBD as reference), IBD dysbiosis state (with non-dysbiotic as reference), age, and antibiotic use using MaAsLin 2's default multivariable linear mixed effects model with subject and site as random effects (**S3 Text**). Features are sorted by minimum FDR-adjusted p-values. For each feature, coefficient estimates and test statistics and the associated two-tailed p-values are also reported. Input features and metadata are also provided.
(XLSX)

**S12 Data. MaAsLin 2 associations between HMP2 multi-omics features (pathway RNA/DNA ratios) and covariates.** List of statistically significant associations (FDR<0.25) between pathway RNA/DNA ratios and IBD disease phenotype (with non-IBD as reference), IBD dysbiosis state (with non-dysbiotic as reference), age, and antibiotic use using MaAsLin 2's default multivariable linear mixed effects model with subject and site as random effects (**S3 Text**). Features are sorted by minimum FDR-adjusted p-values. For each feature, coefficient estimates and test statistics and the associated two-tailed p-values are also reported. Input features and metadata are also provided.
(XLSX)

**S13 Data. MaAsLin 2 associations between HMP2 multi-omics features (stratified DNA pathways) and covariates.** List of statistically significant associations (FDR<0.25) between stratified DNA pathways and IBD disease phenotype (with non-IBD as reference), IBD dysbiosis state (with non-dysbiotic as reference), age, and antibiotic use using MaAsLin 2's default multivariable linear mixed effects model with subject and site as random effects (**S3 Text**). Features are sorted by minimum FDR-adjusted p-values. For each feature, coefficient estimates and test statistics and the associated two-tailed p-values are also reported. Input features and

metadata are also provided.
(XLSX)

**S14 Data. MaAsLin 2 associations between HMP2 multi-omics features (stratified RNA pathways) and covariates.** List of statistically significant associations (FDR<0.25) between stratified RNA pathways and IBD disease phenotype (with non-IBD as reference), IBD dysbiosis state (with non-dysbiotic as reference), age, and antibiotic use using MaAsLin 2's default multivariable linear mixed effects model with subject and site as random effects (**S3 Text**). Features are sorted by minimum FDR-adjusted p-values. For each feature, coefficient estimates and test statistics and the associated two-tailed p-values are also reported. Input features and metadata are also provided.
(XLSX)

**S1 Fig. Details of simulation parameters, evaluation metrics, and benchmarking methods. A)** Four broad metadata designs commonly encountered in microbiome epidemiology for varying sample size, effect size, and feature dimensions are considered: UVA (Single continuous metadata), UVB (Single binary metadata), MVA (Multiple independent metadata), and MVB (Multiple correlated metadata). For each of this broad metadata design, both cross-sectional and longitudinal cases are evaluated (**S1 Text**). **B)** Three aspects of performance are considered: (i) false discovery, (ii) sensitivity, and (iii) scope and computational efficiency of the associated software, each comprising multiple evaluation metrics (**S2 Text**). **C)** A combination of statistical models, normalization, and transformation schemes are employed to the synthetic datasets for a variety of association methods, leading up to 84 combinations of normalization/transformation, zero-inflation, and regression models.
(TIFF)

**S2 Fig. Full summary of detection performance for varying effect size, sample size, and feature dimensions in the simple case of univariate binary metadatum (UVB) without repeated measures.** Both sensitivity and false discovery rates (FDR) are shown for the best-performing methods from each class of methods (as measured by average F1 score). Values are averages over 100 iterations for each parameter combination. The x-axis (effect size) within each panel represents the linear effect size parameter; a higher effect size represents a stronger association. For visualization purposes, only the best-performing methods from each class of models (as measured by average F1 score) are shown. Red line parallel to the x-axis is the target threshold for FDR in multiple testing. Methods are sorted by increasing order of average F1 score across all simulation parameters in this setting. All methods were parallelized using custom bash scripts in a high-performance computing environment and methods unable to process specific simulation configurations due to high computational overhead or slow convergence were omitted for those cases.
(TIFF)

**S3 Fig. Meta-summary of detection performance in the simple case of univariate binary metadatum (UVB) without repeated measures.** Detection performance measures (Sensitivity, FPR, FDR) for all methods are provided. Values are averages over all parameter combinations each summarized over 100 iterations. Red line parallel to the x-axis is the target threshold for FDR in multiple testing. Methods are sorted by increasing order of average F1 score across all simulation parameters in this setting.
(TIFF)

**S4 Fig. Meta-summary of p-value calibration performance in the simple case of univariate binary metadatum (UVB) without repeated measures.** P-value calibration measures as

measured by 'departure from uniformity' (Liberal Area, Conservative Area, Total Area; **S2 Text**) for all methods are displayed. Values are averages over all parameter combinations each summarized over 100 iterations. Red line parallel to the x-axis is the target threshold for FDR in multiple testing. Methods are sorted by increasing order of average F1 score across all simulation parameters in this setting.
(TIFF)

**S5 Fig. Full summary of detection performance for varying effect size, sample size, and feature dimensions in the simple case of univariate continuous metadatum (UVA) without repeated measures.** Both sensitivity and false discovery rates (FDR) are shown for the best-performing methods from each class of methods (as measured by average F1 score). Values are averages over 100 iterations for each parameter combination. The x-axis (effect size) within each panel represents the linear effect size parameter; a higher effect size represents a stronger association. For visualization purposes, only the best-performing methods from each class of models (as measured by average F1 score) are shown. Red line parallel to the x-axis is the target threshold for FDR in multiple testing. Methods are sorted by increasing order of average F1 score across all simulation parameters in this setting. All methods were parallelized using custom bash scripts in a high-performance computing environment and methods unable to process specific simulation configurations due to high computational overhead or slow convergence were omitted for those cases.
(TIFF)

**S6 Fig. Meta-summary of detection performance in the presence of multiple independent metadata (MVA) without repeated measures.** Detection performance measures (F1 score, Matthew's correlation coefficient, FDR) for all methods are displayed. Values are averages over all parameter combinations each summarized over 100 iterations. Red line parallel to the x-axis is the target threshold for FDR in multiple testing. Methods are sorted by increasing order of average F1 score across all simulation parameters in this setting.
(TIFF)

**S7 Fig. Full summary of detection performance for varying effect size, sample size, and feature dimensions in the presence of multiple correlated metadata (MVB) without repeated measures.** Both sensitivity and false discovery rates (FDR) are shown for the best-performing methods from each class of methods (as measured by average F1 score). Values are averages over 100 iterations for each parameter combination. The x-axis (effect size) within each panel represents the linear effect size parameter; a higher effect size represents a stronger association. For visualization purposes, only the best-performing methods from each class of models (as measured by average F1 score) are shown. Red line parallel to the x-axis is the target threshold for FDR in multiple testing. Methods are sorted by increasing order of average F1 score across all simulation parameters in this setting. All methods were parallelized using custom bash scripts in a high-performance computing environment and methods unable to process specific simulation configurations due to high computational overhead or slow convergence were omitted for those cases.
(TIFF)

**S8 Fig. Meta-summary of detection performance in the simple case of univariate binary metadatum (UVB) with repeated measures.** Detection performance measures (Sensitivity, FPR, FDR) for all methods are displayed. Values are averages over all parameter combinations each summarized over 100 iterations. Red line parallel to the x-axis is the target threshold for FDR in multiple testing. Methods are sorted by increasing order of average F1 score across all

simulation parameters in this setting.
(TIFF)

**S9 Fig. Meta-summary of detection performance in the presence of multiple independent metadata (MVA) with repeated measures.** Detection performance measures (Sensitivity, FPR, FDR) for all methods are displayed. Values are averages over all parameter combinations each summarized over 100 iterations. Red line parallel to the x-axis is the target threshold for FDR in multiple testing. Methods are sorted by increasing order of average F1 score across all simulation parameters in this setting.
(TIFF)

**S10 Fig. Meta-summary of detection performance in the simple case of univariate continuous metadatum (UVA) without repeated measures.** Detection performance measures (Sensitivity, FPR, FDR) for all methods are displayed. Values are averages over all parameter combinations each summarized over 100 iterations. Red line parallel to the x-axis is the target threshold for FDR in multiple testing. Methods are sorted by increasing order of average F1 score across all simulation parameters in this setting.
(TIFF)

**S11 Fig. Meta-summary of detection performance in the presence of multiple correlated metadata (MVB) with repeated measures.** Detection performance measures (Sensitivity, FPR, FDR) for all methods are displayed. Values are averages over all parameter combinations each summarized over 100 iterations. Red line parallel to the x-axis is the target threshold for FDR in multiple testing. Methods are sorted by increasing order of average F1 score across all simulation parameters in this setting.
(TIFF)

**S12 Fig. Runtime of association methods.** CPU time (in minutes) is shown for all models faceted by feature dimension (100, 200, 500) and colored by metadata design (i.e., univariate and multivariable) in both cross-sectional (top) and longitudinal (bottom) settings. Values are averages over 100 iterations for each parameter combination. All methods were parallelized using custom bash scripts in a high-performance computing environment and methods unable to process specific simulation configurations due to high computational overhead or slow convergence were omitted for those cases.
(TIFF)

**S13 Fig. Performance of multivariable association methods on negative training data.** MaAsLin 2's default linear model produced a consistently lower proportion of significant associations on negative training data (or repeatedly shuffled training set) (averaged over 1,000 permutations) than the positive training (unshuffled) counterpart in both baseline and longitudinal models (**S3 Text**). Values are average percentages of statistically significant associations (unadjusted $P < 0.05$) summarized over 1000 permutations. Dashed line parallel to the y-axis is the desired 5% significance threshold.
(TIFF)

**S14 Fig. Statistically significant overlap of detected features by MaAsLin 2 and those found in the original study.** Contingency tables describing the intersection of detected features (across all covariates, restricted to common associations found by both methods) between MaAsLin 2 and the original study for various data modalities in the IBDMDB dataset (**S3 Text**).
(TIFF)

**S15 Fig. Overlap of detected dysbiotic taxonomic features by various MaAsLin models.**
Upset plot describing the intersection of detected dysbiotic taxonomic features between various MaAsLin 2 models in the IBDMDB dataset reveals significant overlap across methods (restricted to common associations found by all methods). A similar pattern was observed for functional profiles (data not shown).
(TIFF)

## Author Contributions

**Conceptualization:** Himel Mallick, Timothy L. Tickle, Curtis Huttenhower.

**Data curation:** Himel Mallick, Siyuan Ma, Yancong Zhang, Boyu Ren.

**Formal analysis:** Himel Mallick, Siyuan Ma, Yancong Zhang, Boyu Ren.

**Funding acquisition:** Curtis Huttenhower.

**Investigation:** Himel Mallick, Ali Rahnavard, Lauren J. McIver, Siyuan Ma, Yancong Zhang, Long H. Nguyen, Timothy L. Tickle, George Weingart, Boyu Ren, Emma H. Schwager, Suvo Chatterjee, Kelsey N. Thompson, Jeremy E. Wilkinson, Ayshwarya Subramanian, Yiren Lu, Levi Waldron, Joseph N. Paulson, Eric A. Franzosa, Hector Corrada Bravo, Curtis Huttenhower.

**Methodology:** Himel Mallick, Siyuan Ma, Timothy L. Tickle, Boyu Ren.

**Project administration:** Curtis Huttenhower.

**Resources:** Hector Corrada Bravo, Curtis Huttenhower.

**Software:** Himel Mallick, Ali Rahnavard, Lauren J. McIver.

**Supervision:** Eric A. Franzosa, Hector Corrada Bravo, Curtis Huttenhower.

**Validation:** Himel Mallick, Ali Rahnavard, Lauren J. McIver, Siyuan Ma, Timothy L. Tickle, Boyu Ren, Emma H. Schwager, Suvo Chatterjee.

**Visualization:** Himel Mallick, Ali Rahnavard, Lauren J. McIver.

**Writing – original draft:** Himel Mallick, Ali Rahnavard, Lauren J. McIver, Siyuan Ma, Yancong Zhang, Long H. Nguyen, Timothy L. Tickle, George Weingart, Boyu Ren, Emma H. Schwager, Suvo Chatterjee, Kelsey N. Thompson, Jeremy E. Wilkinson, Ayshwarya Subramanian, Yiren Lu, Levi Waldron, Joseph N. Paulson, Eric A. Franzosa, Hector Corrada Bravo, Curtis Huttenhower.

**Writing – review & editing:** Himel Mallick, Ali Rahnavard, Lauren J. McIver, Siyuan Ma, Yancong Zhang, Long H. Nguyen, Timothy L. Tickle, George Weingart, Boyu Ren, Emma H. Schwager, Suvo Chatterjee, Kelsey N. Thompson, Jeremy E. Wilkinson, Ayshwarya Subramanian, Yiren Lu, Levi Waldron, Joseph N. Paulson, Eric A. Franzosa, Hector Corrada Bravo, Curtis Huttenhower.

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
