## [Editor Report · Decision Letter 0]

9 Sep 2021

Dear Dr. Huttenhower,

We are pleased to inform you that your manuscript 'Multivariable Association Discovery in Population-scale Meta-omics Studies' has been provisionally accepted for publication in PLOS Computational Biology.

Best regards,

Luis Pedro Pedro Coelho

Associate Editor

PLOS Computational Biology

Jian Ma

Deputy Editor

PLOS Computational Biology

Reviewers #1 and #2 both agreed on the value of the manuscript. Their concerns focused on matters of presentation or acknowledgement of limitations, which the authors have now addressed.

Reviewer #3 raised concerns related to the novelty of the work (which reviewer #2, whilst not questioning the manuscript as a whole, also alludes to in their comment on the comparison with MaAsLin1). However, taking the authors' response into account as well as the fact that the manuscript is now presented as a Software Paper, I feel that there is sufficient novelty to warrant publication. For a long time, users of MaAsLin (of which there are many) were in the awkward position of having to cite a somewhat unrelated paper as a reference as a proper benchmarking of the tool was lacking. Thus, in the absence of reviewer concerns of a technical nature, I believe this paper will fill a gap in the literature.

---

## [Editor Report · Acceptance letter]

11 Nov 2021

PCOMPBIOL-D-21-01441 

Multivariable Association Discovery in Population-scale Meta-omics Studies

Dear Dr Huttenhower,

I am pleased to inform you that your manuscript has been formally accepted for publication in PLOS Computational Biology. Your manuscript is now with our production department and you will be notified of the publication date in due course.

With kind regards,

Livia Horvath
